# Endocrine and Metabolic Illnesses in Young Adults with Prader–Willi Syndrome

**DOI:** 10.3390/jpm12060858

**Published:** 2022-05-25

**Authors:** Eu-Seon Noh, Min-Sun Kim, Chiwoo Kim, Kyeongman Jeon, Seonwoo Kim, Sung Yoon Cho, Dong-Kyu Jin

**Affiliations:** 1Department of Pediatrics, Samsung Medical Center, Sungkyunkwan University School of Medicine, Seoul 06351, Korea; aa.noh@samsung.com (E.-S.N.); min-sun.kim@samsung.com (M.-S.K.); chngkak@gmail.com (C.K.); jindk.jin@samsung.com (D.-K.J.); 2Division of Pulmonary and Critical Care Medicine, Department of Medicine, Samsung Medical Center, Sungkyunkwan University School of Medicine, Seoul 06351, Korea; kyeongman.jeon@samsung.com; 3Academic Research Service Headquarter, LSK Global PS, Seoul 04535, Korea; kim.seonwoo@lskglobal.com

**Keywords:** Prader–Willi syndrome, endocrine, young adult, obesity, diabetes mellitus, dyslipidemia, decreased bone density, sleep apnea

## Abstract

Prader–Willi syndrome (PWS) is a rare genetic disorder characterized by an insatiable appetite that leads to morbid obesity. Previous studies reported health problems in adults with PWS. However, studies on younger adults are lacking, and there are no specific studies of endocrine and metabolic illness in this age group. We performed a retrospective cohort study of 68 individuals with PWS aged 19 to 34 years at Samsung Medical Center. The prevalence of endocrine and metabolic illnesses were compared with those in an age-, sex-, and BMI-matched healthy control group. Young adults with PWS had a higher prevalence of metabolic syndrome (35.3% vs. 4.4%), type 2 diabetes mellitus (50.0% vs. 5.4%), hypertension (30.8% vs. 16.1%), dyslipidemia (38.2% vs. 14.7%), decreased bone density (26.4% vs. 0.9%), and sleep apnea (32.3% vs. 4.4%) than controls (all *p* < 0.05). The PWS group that maintained recombinant human growth (rhGH) treatment in adulthood had a lower probability of having a BMI ≥ 30 at the last follow-up (odds ratio = 0.106 (0.012–0.948), *p* = 0.045). Endocrine and metabolic illnesses in individuals with PWS may have already started in the early teens; therefore, appropriate screening and early intervention are important. Better understanding of the natural history of PWS and age-related complications will lead to better-quality medical care for individuals with PWS.

## 1. Introduction

Prader–Willi syndrome (PWS) is a rare genetic disorder characterized by muscular hypotonia, dysmorphic features, short stature, low lean body mass, mental retardation, behavioral abnormalities, and progressive obesity owing to hyperphagia [1]. The benefits of recombinant human growth hormone (rhGH) treatment with children with PWS on body composition, muscle characteristics, sleep disordered breathing, and metabolic parameters have been established [2]. In adults with PWS, rhGH treatment may have a beneficial effect on body composition by increasing lean body mass and decreasing fat mass [3]. However, there have been some concerns about the safety of rhGH treatment in adults with PWS [4]. With improvements in diagnosis and medical care, individuals with PWS live longer and more often survive to adulthood, but metabolic and endocrine illnesses occur more frequently in these individuals during the aging process than the general population, largely due to severe obesity. A few studies have investigated health problems in adults with PWS [5,6]. However, most of them included adult patients with a wide age range, and no studies have specifically focused on endocrine and metabolic illnesses in young adults with PWS. Young adulthood is an important transition period from dependent to independent, and the development of obesity has not been well studied [7]. In addition, transitional health care is considered a very important issue due to the many physical and psychological changes that occur during transitions. Here, we investigated endocrine and metabolic illnesses in young adults with PWS in Korea and compared them with age-, sex-, and BMI-matched healthy controls. In addition, we evaluated the association of rhGH treatment for each one of the endocrine and metabolic illnesses in young adults with PWS. 

## 2. Materials and Methods

### 2.1. Patients

We reviewed the medical records of adults with PWS diagnosed by clinical criteria and methylation PCR between March 1994 and May 2021 at Samsung Medical Center. Adults aged 19 to 34 years at the last follow-up were selected; 68 subjects were included in this study. All subjects were Korean individuals with PWS who visited our hospital on a six-monthly basis. 

To compare the prevalence of endocrine and metabolic illnesses, we collected a healthy control group with matching age-, sex-, and BMI groups. A total of 204 individuals (aged 19–34 years) were selected from among patients who visited the Department of Family Medicine or the Examination Center. None of these individuals had genetic syndromes or chronic disorders. 

In addition, the group with history of rhGH treatment, the group that continued to receive rhGH treatment after adulthood, and the group that had never received rhGH treatment were compared within our PWS cohort.

### 2.2. Data Collection and Measurements

Medical and biometric data, including height, weight, vital signs, physical examination findings, blood tests, spine X-ray findings, dual energy X-ray absorptiometry (DXA) findings, and polysomnography (PSG) findings were collected from medical records. Subjects were categorized into one of five groups according to body mass index (BMI) following the World Health Organization’s recommendations for Asians: normal weight, 18.5–22.9 kg/m^2^; overweight, 23.0–24.9 kg/m^2^; obese class I, 25.0–29.9 kg/m^2^; obese class II, 30.0–34.9 kg/m^2^; obese class III ≥ 35 kg/m^2^. Blood samples were collected in the morning in the fasting state for the measurement (in serum) of lipids, glucose, insulin, hemoglobin A1c (HbA1C), free T4, T3, TSH, ACTH, and cortisol. All laboratory data were analyzed at Samsung Medical Center.

### 2.3. Definitions of Endocrine and Metabolic Illnesses

ADA guidelines were used to diagnose diabetes mellitus. Hypertension was defined as a systolic blood pressure (SBP) ≥ 140 mmHg or diastolic blood pressure (DBP) ≥ 90 mmHg or on an anti-hypertensive medication. Dyslipidemia was defined as taking a medication for uncontrolled lipid abnormalities despite lifestyle changes. Individuals who met any of the following lipid abnormalities were defined as having dyslipidemia: (1) Triglycerides ≥ 200 mg/dL, (2) LDL-C ≥ 160 mg/dL, and (3) HDL-C ≤ 40 mg/dL [8].

Metabolic syndrome was defined as a requirement for fasting glucose ≥110 mg/dL (impaired fasting glucose or diabetes mellitus), plus at least two of the following: (i) obesity, defined as a BMI; ≥25 kg/m^2^; (ii) dyslipidemia; and (iii) hypertension or on an anti-hypertensive medication.

Primary hypothyroidism was defined as a TSH level ≥ 10 μIU/mL with a free T4 level of <0.75 ng/dL. Central hypothyroidism was defined as a free T4 level of <0.75 ng/dL and non-elevated TSH levels. Secondary hypothyroidism was indicated by a low basal serum free T4 level with an inappropriately normal or low TSH level (not increased by >5 mU/L) in a thyrotropin-releasing hormone stimulation test. Hyperthyroidism was defined as a TSH level of ≤0.10 μIU/mL and elevated free T3 and/or free T4 levels. 

Decreased bone density was defined as a T-score (comparison of a person’s bone density with that of a healthy 30-year-old of the same sex) below −2.5 on DXA in patients who underwent DXA at 19 years or older than 19 years, and as a Z-score (adjusted by age- and gender-matched standards for Korean children) below −2.0 in patients who underwent DXA when younger than 19 years of age [9].

PSG was performed in patients who showed symptoms suspicious of sleep apnea and who were cooperative when conducting the test. Sleep apnea was classified based on the apnea–hypopnea index, i.e., the mean number of apneas and hypopneas per hour of sleep. An apnea–hypopnea index of 5–15 per hour was classified as mild sleep apnea, 15–30 per hour as moderate sleep apnea, and >30 per hour as severe sleep apnea.

The presence of central adrenal insufficiency was investigated by estimation of serum ACTH and cortisol levels in blood samples obtained in the early morning and by low-dose ACTH stimulation test, when needed. The cutoff level for appropriate cortisol response was accepted as 18 mcg/dL. 

### 2.4. Growth Hormone Treatment

In our cohort, the dose of rhGH in patients receiving rhGH treatment was 1.0 mg/m^2^ per day (not exceeding a maximum of 2.7 mg per day). After reaching the final height or epiphyseal closure, if growth hormone deficiency was confirmed through growth hormone provocation test, the treatment for adult growth hormone deficiency was maintained. Adult rhGH doses ranged from 0.4 to 0.5 mg per day [4,10]. rhGH treatment exclusions for in patients with PWS included severe obesity, uncontrolled diabetes, untreated severe obstructive sleep apnea, active psychosis, and lack of compliance from parents.

### 2.5. Statistical Analyses

Baseline characteristics are presented as the number of frequency (percentage) for categorical variables and the mean ± SD and range for continuous variables. Chi-square test and Fisher’s exact test were used to determine the significance of differences in the proportions of endocrine and metabolic illnesses between the PWS and control group. Comparisons between groups divided by rhGH treatment status were performed using Fisher’s exact test for categorical variables and one-way ANOVA for continuous variables. Crude and adjusted logistic regression analyses were performed to assess the associations between rhGH treatment and each one of the endocrine and metabolic illnesses. All statistical analyses were performed using SPSS 27 (IBM Corp., Armonk, NY, USA). A two-sided *p* value < 0.05 was considered statistically significant. 

## 3. Results

### 3.1. Clinical Characteristics of Young Adults with PWS

Demographic and clinical characteristics are presented in Table 1. Mean age at last follow-up was 24.5 (SD 4.2) years. Mean age at diagnosis of PWS was 6.9 (SD 5.7) years. Forty-seven (69.1%) patients had deletion type PWS. Mean BMI was 34.6 (SD 11.7). Fifty-eight patients (85.2%) underwent GH treatment. Two patients died in our cohort. One of these patients died suddenly at home at the age of 31 years. This male was taking medications for type 2 diabetes mellitus (T2DM) and hypertension and was prescribed continuous positive airway pressure (CPAP) due to severe sleep apnea but showed poor compliance. The patient’s last BMI was 56.8. The other patient died at age 29 due to pneumonia and exacerbation of pulmonary hypertension. This female had suffered from uncontrolled T2DM despite insulin treatment since age 20. She was also on diuretics and oxygen therapy since 25 years old due to pulmonary hypertension and right ventricular (RV) dysfunction. Her last BMI was 44.97. She was being treated with CPAP for severe sleep apnea and had used a formoterol and budesonide inhaler for asthma since age 27.

### 3.2. Comparison of Health Problems between PWS Cohort and Age-, Sex-, and BMI-Matched Healthy Controls

Young adults with PWS had a higher prevalence of metabolic syndrome, T2DM, hypertension, dyslipidemia, decreased bone density, and sleep apnea than the age-, sex-, and BMI-matched healthy controls (*p* < 0.05), but there was no significant difference in thyroid disorders between the two groups (Table 2).

### 3.3. Age at Diagnosis of Endocrine and Metabolic Illnesses

The distribution of the age at diagnosis of endocrine and metabolic illnesses in young adults with PWS is shown in Figure 1. T2DM, dyslipidemia, thyroid disorder, decreased bone density, and sleep apnea were diagnosed in the late teens, on average, while hypertension was diagnosed in the early twenties.

### 3.4. Type 2 Diabetes Mellitus 

Thirty-four patients were diagnosed with T2DM at the mean age of 16.61 (SD 3.73) years. Table 3 provides a summary of the comorbidities and antidiabetic drugs prescribed to our cohort of young adults with PWS. The most frequent comorbidity was hypertension (61.7%), followed by diabetic nephropathy (47.0%) and diabetic neuropathy (17.6%). One patient had retinopathy, and two patients had DM foot; all of these patients had poor glucose control. Oral hypoglycemic agents (OHAs) were prescribed to 32 (94.1%) patients, while a combination of OHA and insulin was prescribed to 65.6% of patients. Of the patients on OHAs alone, six were on combination therapy, while five were receiving monotherapy. The most common combination therapy was metformin and sulfonylureas. The OHA used for monotherapy was metformin. Eleven patients were treated with a combination of a glucagon-like peptide 1 (GLP-1) receptor agonist and metformin, of which nine were also treated with insulin.

### 3.5. Hypertension

Hypertension was diagnosed in 21 of the 68 patients. Mean age at diagnosis of hypertension was 22.20 (SD 4.24) years. Medications included calcium-channel blockers (five patients), angiotensin II receptor blockers (seven patients), and angiotensin converting enzyme inhibitors (nine patients).

### 3.6. Dyslipidemia

Twenty-six patients took medications for hyperlipidemia. Mean age when medications to treat this were started was 17.74 (SD 3.83) years. There were 16 patients with combined hyperlipidemia, 8 patients with isolated hypercholesterolemia, and 2 patients with isolated hypertriglyceridemia. Among these patients, 11 patients had hypoalphalipoproteinemia. Twenty-four patients took an HMG-CoA reductase inhibitor, and one patient also took omega-3 fatty acids. The other two patients took only omega-3 fatty acids.

### 3.7. Thyroid Disorders

Most patients in our cohort had normal thyroid function. Two females had primary hypothyroidism with TSH levels ≥ 10 μIU/mL and free T4 levels < 0.75 ng/dL at diagnosis age of 19 years and 14 years, respectively. Their thyroid antibody test was negative. There was no hyperthyroidism in our cohort.

### 3.8. Decreased Bone Density 

DXA was performed in 41 patients. Eight patients (26.4%) had decreased bone density. Mean age at diagnosis of decreased bone density was 15.18 (SD 3.36) years. Among patients with decreased bone density, four males had hypogonadism, but hormone replacement was not performed due to concerns about possible behavioral-related side effects.

### 3.9. Sleep Apnea

PSG was performed in 34 patients. Sleep apnea was confirmed in 22 patients, and the mean age at diagnosis of sleep apnea was 15.35 (SD 6.59) years old. Mean total sleep apnea–hypopnea index was 26.94 (SD 29.57) per hour. Six of the patients with sleep apnea received CPAP treatment. One of these patients was treated with uvulo-palato-pharyngoplasty, radiofrequency tongue base volume reduction, and palatine tonsillectomy due to severe sleep apnea. Echocardiography was performed in 10 patients with sleep apnea. Among them, one patient had an increase in right atrial pressure up to 15 mmHg and RV dilatation, while another patient had decreased RV systolic function and severe pulmonary hypertension. Patients with sleep apnea had a significantly higher BMI than patients without sleep apnea (42.99 (SD 14.15) vs. 30.73 (SD 7.89), respectively, *p* < 0.05).

### 3.10. Adrenal Insufficiency

Adrenal function was evaluated in 30 patients (44.1%); all of them were found to have sufficient hypothalamic–pituitary–adrenal axis function and no indication for perioperative hydrocortisone treatment.

### 3.11. Growth Hormone Treatment in Young Adults with PWS

In our study, 58 patients were treated with rhGH since childhood. Of these, 48 patients discontinued treatment at a mean age of 14 (SD 2.4) years, and 10 patients continued to receive adult rhGH treatment. One patient was diagnosed with PWS at the age of 21 and was subsequently treated with adult rhGH treatment. Among the patients treated with rhGH treatment, six patients discontinued rhGH treatment prematurely due to adverse events. Four patients discontinued due to uncontrolled hyperglycemia, and two patients discontinued due to worsening of the scoliosis.

A comparison of demographic, metabolic and endocrine illnesses in young adults with PWS according to rhGH treatment status and comparison of age-, sex-, and BMI-matched healthy control group with each group are shown in Table 4a. In the non-rhGH treatment group, the BMI at the last follow-up was 41.65 (SD 14.78), and in the history of rhGH treatment group, it was 32.58 (SD 11.1), and in the currently rhGH treatment group, it was 37.8 (SD 8.7), and this difference was statistically significant. When compared with the control group with matched age, sex, and BMI and each group divided according to rhGH treatment status, endocrine and metabolic diseases still had a significantly higher prevalence in the PWS group regardless of the history of rhGH treatment, except for thyroid disease and hypertension. 

Crude and adjusted logistic regression analyses were performed to assess the associations between rhGH treatment status and each one of the endocrine and metabolic illnesses (Table 4b). Thyroid disorders were excluded because they were not acceptable to the regression model. They were not associated with the rhGH treatment and outcome variables such as metabolic syndrome, diabetes, hypertension, dyslipidemia, decreased bone density, and sleep apnea. However, the PWS group maintaining rhGH treatment in adulthood was inversely associated with having BMI ≥ 30 at the last follow-up (odds ratio = 0.106 (0.012–0.948), *p* = 0.045).

## 4. Discussion

This is the first study to investigate endocrine and metabolic illnesses in adults with PWS in Korea. In addition, this is the first study to focus on young adults with PWS worldwide. In this study, young adults with PWS had a higher prevalence of metabolic syndrome, T2DM, hypertension, dyslipidemia, decreased bone density, and sleep apnea than an age-, sex-, and BMI-matched healthy control group. Mean age at diagnosis of these problems ranged from the late teens to the early twenties, emphasizing the importance of appropriate screening and intervention during this period. The high morbidity and mortality rates associated with PWS are related to severe obesity as a result of hyperphagia [11]. Therefore, intervention for hyperphagia to prevent severe obesity and early diagnosis and management of complications related to obesity are important to improve the prognosis of these patients.

Severe obesity is a significant risk factor for developing T2DM in individuals with PWS as in other populations. Perturbations in glucose metabolism occur in PWS individuals during the transition phase to adulthood [12,13]. In our cohort, T2DM was diagnosed in half of the 68 patients. PWS patients had a higher prevalence of T2DM than the healthy control group. In a previous cohort study of subjects with PWS over 10 years of age at our center, the prevalence of T2DM was 13% [14] but increased to 50.0% when young adults over 19 years of age were included in the study.

Difficulties in compliance with diet and food compulsivity can explain the severe uncontrolled diabetes in some patients. In our study, there were more cases of glucose control with a combination therapy with insulin or other oral hypoglycemic medications than with metformin monotherapy. Furthermore, 10 patients were treated with a combination therapy of a glucagon-like peptide 1 (GLP-1) receptor agonist due to uncontrolled diabetes. A recent systematic review has shown that GLP-1 receptor agonists are safe in patients with PWS and may potentially help with weight reduction, blood sugar control, and appetite control [15]. In our cohort, GLP-1 receptor agonists appeared to be well tolerated with no serious adverse effects; however, four patients discontinued the injections because their parents did not feel they had a significant effect on appetite, although these patients had a slightly lowered Hba1c during the early phase of the injection. Poor adherence with injections may pose a major obstacle to maintenance therapy in these patients. Randomized, controlled, long-term clinical trials of the efficacy and safety of GLP1 receptor agonists in patients with PWS of various age groups are needed. 

Hypertension and dyslipidemia were highly prevalent in our cohort, with prevalence rates higher than those reported previously [6,16]. Older age has also been found to be associated with hypertension and dyslipidemia, probably due to severe obesity in older patients [17]. In our center, regular follow-up of screening of metabolic syndrome is performed at least every 6 months; therefore, early diagnosis of these problems is possible compared to previous studies. In addition, the PWS cohort had a higher prevalence of hypertension and dyslipidemia than the control group, likely due to the effect of an extremely high BMI and difficulty with lifestyle modifications owing to behavioral problems in the PWS cohort.

Hypothyroidism was found in only two (2.9%) patients in our cohort, which was not a significant difference in frequency from that in the healthy control group. Thyroid axis dysfunction is a frequent feature during infancy in individuals with PWS [18]. Therefore, screening for hypothyroidism within the first 3 months of life and then yearly is recommended. In our cohort, six patients showed a transient elevation in TSH only when they were infants.

Both low BMD and increased fracture risk have been described in adults with PWS [19]. Our study revealed a low BMD in young adults with PWS compared to the healthy control group. There was no difference in the prevalence of decreased bone density between the GH-treated group and the non-GH group. A previous report suggested that sex hormones play a major role in conservation of BMD in adolescents and adults with PWS. Sex hormone replacement therapy (HRT) should be considered in non-gonadal adolescents with PWS to improve bone health in adulthood [20]. However, behavioral problems are a serious issue in PWS and can make guardians and medical staff reluctant to start HRT. 

The higher risk of central sleep apnea in individuals with PWS is associated with hypothalamic dysfunction [21]. However, several other factors, such as obesity, hypotonia, scoliosis, narrowing of the upper airways, reduction in saliva excretion, and adenoid/tonsillar hypertrophy, can also lead to obstructive sleep apnea (OSA) [22]. Sleep apnea is associated with increased sympathetic activation, vagal withdrawal, altered hemodynamic loading conditions, and hypoxemia. Moreover, OSA is strongly associated with arterial hypertension, the most common risk factor for cardiac hypertrophy and failure. CPAP treatment is currently the best treatment option for OSA in patients with heart failure [23]. In our cohort, 22 (32.3%) patients showed mixed central sleep apnea and OSA. Although CPAP treatment was recommended for these 22 patients, only 6 patients used CPAP. Uvulo-palato-pharyngoplasty, radiofrequency tongue base volume reduction, and palatine tonsillectomy was performed in one patient with severe apnea. Although adeno-tonsillectomy often does not cure OSA in PWS, it can improve OSA severity and quality of life [24]. The high prevalence of complications after surgery and residual OSA should be addressed in perioperative counseling. In our cohort, there was no difference in sleep apnea prevalence between the GH-treated group and the non-GH group. However, the development of OSA has been reported following GH treatment in some children with PWS [25]. A repeat sleep study should be performed 3 to 6 months after starting GH treatment, and patients should continue to be monitored for signs and symptoms of OSA thereafter. Weight management is important because higher BMI increases the risk of sleep apnea.

Central adrenal insufficiency can be a part of PWS. Clinicians need an evaluation of adrenal function in patients with PWS. However, the frequency of central adrenal insufficiency in patients with PWS is not clear, and the frequency varies widely between studies [26,27,28]. In our cohort, none of them had central adrenal insufficiency. Although central adrenal insufficiency is rare in adults with PWS [27], administration of hydrocortisone during illness or surgery should be individualized. 

rhGH treatment has become the standard of care in PWS children, even in those who do not have proven GH deficiency [4]. rhGH treatment causes improvements in physical health and cognition and might improve quality of life [29]. Several systematic reviews on rhGH treatment in adults with PWS have been published in recent years. However, there have been some concerns regarding the safety of rhGH treatment in adults with PWS [4]. The most recent meta-analysis showed rhGH treatment improves body composition in adults with PWS, without safety concerns. Furthermore, it may improve muscle strength, endurance, metabolic health, cognition, and quality of life [30]. Similarly to previous findings, we observed that the BMI at the last follow-up was significantly lower in the history of rhGH treatment group and currently rhGH treatment group than non-rhGH treatment group. In addition, the currently rhGH treatment group had a lower probability of having a BMI ≥ 30 at the last follow-up. However, that BMI is of limited value in assessing body composition in patients with PWS because of the extreme relationship between body fat and lean body mass. Additionally, the effect size of the benefits is small, and further longer-term controlled studies on the benefits and risks of rhGH treatment in adult in this patient population are necessary.

In our cohort, 11 adult patients were receiving adult rhGH treatment. Only one patient discontinued rhGH treatment due to uncontrolled hyperglycemia. The rhGH treatment was not associated with the prevalence of endocrine and metabolic illnesses such as metabolic syndrome, diabetes, hypertension, dyslipidemia, decreased bone density, and sleep apnea. Additionally, rhGH treatment is generally well tolerated, and there were no major safety issues in young adult patients in our PWS cohort. If rhGH is used in adults with PWS, this should be managed by a specialist multidisciplinary team.

The annual mortality rate of adults with PWS is approximately 3%, and the average age at death is 33 years. Causes of death of adolescents and adults with PWS are related to cardiac disease and failure, pulmonary thromboembolism, accidents, sepsis, and obesity-related complications. In contrast, respiratory failure, aspiration, infection, and choking are causes of death in infants and young children with PWS [31]. Early intervention to prevent severe obesity is important to reduce the mortality of individuals with PWS. [32]

This study had several limitations. This was a retrospective study, and there were missing values related to decreased bone density, sleep apnea, and hypogonadism, both of secondary and primary origin. The strengths of our study are our focus on young adults, which allowed us to exclude aging-associated conditions and diseases. We investigated endocrine and metabolic illnesses in this cohort, and our results emphasize the need for specialized guidance and medical care in the transition from childhood to adulthood. In addition, we found that endocrine and metabolic illnesses were more prevalent in young adults with PWS than in a normal age-, sex-, and BMI-matched healthy control group.

## 5. Conclusions

In our study, young adults with PWS were more likely to develop endocrine and metabolic illnesses than the age-, sex-, and BMI-matched healthy controls that likely started in their teen years, highlighting the importance of appropriate screening and intervention at this time. The rhGH treatment was not associated with the prevalence of metabolic syndrome, diabetes, hypertension, dyslipidemia, decreased bone density, and sleep apnea. The PWS group that had rhGH treatment in adulthood had a lower probability of having a BMI ≥ 30 at the last follow-up. The rhGH treatment is generally well tolerated, and there were no major safety issues in our young adult with PWS. Better understanding of the natural history of PWS patients and age-related complications will improve the quality of medical care in adults with PWS. Further studies of behavioral and psychiatric problems and the quality of life of patients and parents with PWS, and longer-term controlled studies of the benefits and risks of rhGH treatment in adult with PWS are necessary. Furthermore, we are hopeful that the development of appetite suppressant drugs will help control obesity and reduce the incidence of related diseases in PWS.

## Figures and Tables

**Figure 1 jpm-12-00858-f001:**
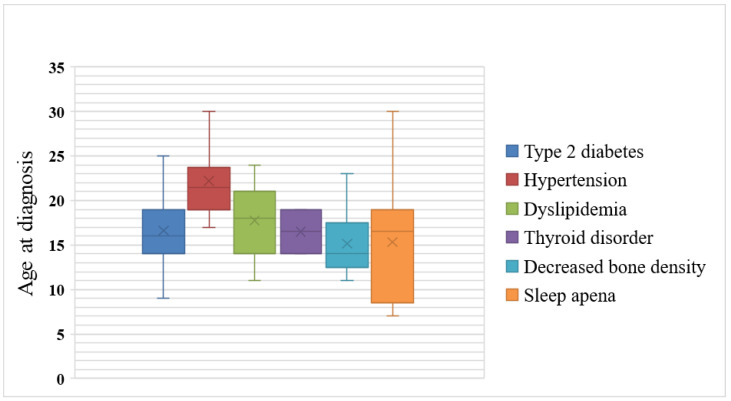
Box and whisker plot of age at diagnosis of endocrine and metabolic illnesses in young adults with Prader–Willi syndrome.

**Table 1 jpm-12-00858-t001:** Demographic and clinical characteristics of 68 young adults with Prader–Willi syndrome.

Young Adult PWS, *n* = 68	
**Age (years) (range)**	24.5 ± 4.2 (19.0~34.0)
**Age at diagnosis with PWS (range)**	6.9 ± 5.7 (0.1~28.5)
**Sex** **(male), *n* (%)**	39 (57.3)
**Genotype (deletion), *n* (%)**	44 (64.7)
**BMI in kg/m^2^**	34.6 ± 11.7 (17.5~78.9)
**<22.9, *n* (%)**	5 (7.3)
**23–24.9, *n* (%)**	6 (8.8)
**25–29.9, *n* (%)**	15 (22.0)
**30–34.9, *n* (%)**	20 (29.4)
**35<, *n* (%)**	22 (32.3)
**Obesity, *n* (%)**	58 (85.2)
**History of growth hormone treatment, *n* (%)**	58 (85.2)
**Death, *n* (%)**	2 (2.9)
**Metabolic syndrome, *n* (%)**	24 (35.3)
**Type 2 diabetes** **, *n* (%)**	34 (50.0)
**Hypertension** **, *n* (%)**	21 (30.8)
**Dyslipidemia** **, *n* (%)**	26 (38.2)
**Thyroid disorder** **, *n* (%)**	2 (2.9)
**Decreased bone density** **, *n* (%)**	18 (26.4)
**Sleep apnea** **, *n* (%)**	22 (32.3)
**Adrenal insufficiency, *n* (%)**	0 (0)
**Scoliosis** **, *n* (%)**	41 (60.2)
**Behavior disorder** **, *n* (%)**	24 (45.2)
**Soft tissue infection** **, *n* (%)**	4 (5.8)

Data are presented as numbers (%) or mean ± SDs.

**Table 2 jpm-12-00858-t002:** Comparison of endocrine and metabolic illnesses in the Prader–Willi syndrome group and the age-, sex-, and BMI-matched healthy control group.

	PWS (*n* = 68)	Control (*n* = 204)	*p*-Value
**Metabolic syndrome, *n* (%)**	24 (35.3)	9 (4.4)	<0.001
**Type 2 diabetes** **, *n* (%)**	34 (50.0)	11 (5.4)	<0.001
**Hypertension** **, *n* (%)**	21 (30.8)	33 (16.1)	0.0085
**Dyslipidemia** **, *n* (%)**	26 (38.2)	29 (14.2)	<0.001
**Thyroid disorder** **, *n* (%)**	2 (2.9)	5 (2.4)	0.5574
**Decreased bone density** **, *n* (%)**	18 (26.4)	2 (0.9)	<0.001
**Sleep apnea** **, *n* (%)**	22 (32.3)	9 (4.4)	<0.001

Data are presented as the number (%).

**Table 3 jpm-12-00858-t003:** Comorbidities and types of antidiabetic drugs prescribed for treatment of type 2 diabetes mellitus in young adults with Prader–Willi syndrome.

	Number (%)
**Comorbidities**	
Diabetic nephropathy	16 (47.0)
Diabetic neuropathy	6 (17.6)
Diabetic retinopathy	1 (2.9)
Hypertension	21 (61.7)
Diabetic foot	2 (5.8)
Heart failure	2 (5.8)
**Antidiabetics**	
OHA alone	11 (32.3)
Combination therapy	6 (17.6)
Monotherapy	5 (14.7)
Insulin alone	0 (0)
OHA and GLP-1 receptor agonist	2 (5.8)
OHA and insulin	12 (35.2)
OHA and insulin and GLP-1 receptor agonist	9 (26.4)

OHA: oral hypoglycemic medications; GLP-1 receptor agonist: glucagon-like peptide 1 receptor agonist.

**Table 4 jpm-12-00858-t004:** (**a**) Demographic comparison. Metabolic and endocrine illnesses in young adults with PWS according to rhGH treatment, and comparison of age-, sex-, and BMI-matched healthy control group with each group. (**b**) Logistic regression model assessing the association of metabolic and endocrine illnesses with rhGH, crude- and age- and sex-adjusted for PWS patients.

**(a)**							
	**Non-rhGH** **T** **reatment (*n* = 10)**		**History of rhGH Treatment (*n* = 48)**		**Currently rhGH** **Treatment (*n* = 10)**		** *p* ** **-Value**
**Age (Years) (Range)**	28.5 ± 4.8 (20–33)		23.5 ± 3.4 (19–31)		23.6 ± 4.3 (19–32)		0.757
**Age of starting rhGH Tx. (years)**			8.2 ± 3.6 (2–16)		9.6 ± 6.0 (3–22.4)		0.350
**Age at diagnosis with PWS (years)**	11.6 ± 9.4 (0.1–28.5)		5.5 ± 4.2 (0.1–13.7)		6.7 ± 5.4 (0.1–22.4)		0.869
**Sex (male), *n* (%)**	8 (80)		26 (54.1)		5 (50)		0.443
**Genotype (deletion), *n* (%)**	6 (60)		30 (62.5)		8 (80)		0.340
**BMI, kg/m^2^ (±SD) (range)**	41.65 ± 14.78 (26–78)		32.58 ± 11.1 (18–58)		37.8 ± 8.7 (25–53)		0.048
		** *p* ** **-Value ***		** *p* ** **-Value ***		** *p* ** **-Value ***	
**Metabolic syndrome, *n* (%)**	5 (50)	<0.001	16 (33.3)	<0.001	3 (30)	0.013	
**Type 2 diabetes** **, *n* (%)**	5 (50)	<0.001	25 (52.1)	<0.001	4 (40)	0.003	
**Hypertension** **, *n* (%)**	6 (60)	0.003	12 (25)	0.112	2 (20)	0.509	
**Dyslipidemia, *n* (%)**	4 (40)	0.05	17 (35.4)	0.001	5 (50)	0.011	
**Thyroid disorder** **, *n* (%)**	0 (0)	0.785	2 (4.2)	0.398	0 (0)	0.785	
**Decreased bone density** **, *n* (%)**	2 (20)	0.011	14 (29.2)	<0.001	2 (20)	0.011	
**Sleep apnea, *n* (%)**	3 (30)	0.013	16 (33.3)	<0.001	3 (30)	0.013	
**(b)**							
**Outcome**	**Variable**	**Crude OR**			**Adjusted OR** ** * ^a^ * **		
		**Coef (95% CI)**		** *p* ** **-Value**	**Coef (95% CI)**		** *p* ** **-Value**
**Metabolic syndrome**	Non-rhGH treatment	Ref		0.571	Ref		0.937
	History of rhGH treatment	2.333 (0.373–14.613)		0.365	1.461 (0.180–11.856)		0.723
	Currently rhGH treatment	1.167 (0.266–5.123)		0.838	1.131 (0.245–5.221)		0.875
**Type 2 diabetes**	Non-rhGH treatment	Ref		0.787	Ref		0.692
	History of rhGH treatment	1.500 (0.255–8.817)		0.654	0.929 (0.121–7.131)		0.943
	Currently rhGH treatment	1.630 (0.408–6.521)		0.489	1.623 (0.383–6.875)		0.511
**Hypertension**	Non-rhGH treatment	Ref		0.091	Ref		0.508
	History of rhGH treatment	6.000 (0.812–44.351)		0.079	3.317 (0.363–30.347)		0.288
	Currently rhGH treatment	1.333 (0.248–7.165)		0.737	1.323 (0.237–7.391)		0.749
**Dyslipidemia**	Non-rhGH treatment	Ref		0.647	Ref		0.447
	History of rhGH treatment	0.429 (0.068–2.684)		0.365	0.258 (0.032–2.094)		0.205
	Currently rhGH treatment	0.600 (0.152–2.362)		0.465	0.599 (0.150–2.396)		0.468
**Decreased bone density**	Non-rhGH treatment	Ref		0.740	Ref		0.854
	History of rhGH treatment	1.000 (0.112–8.947)		1.000	1.374 (0.115–16.384)		0.801
	Currently rhGH treatment	1.647 (0.310–8.748)		0.558	1.622 (0.292–9.018)		0.581
**Sleep apnea**	Non-rhGH treatment	Ref		0.965	Ref		0.690
	History of rhGH treatment	1.000 (0.148–6.772)		1.000	0.555 (0.064–4.818)		0.594
	Currently rhGH treatment	1.167 (0.266–5.123)		0.838	1.209 (0.266–5.493)		0.806
**BMI** **≥** **30**	Non-rhGH treatment	Ref		0.060	Ref		0.127
	History of rhGH treatment	0.444 (0.034–5.880)		0.538	0.152 (0.009–2.643)		0.196
	Currently rhGH treatment	0.121 (0.014–1.029)		0.053	0.106 (0.012–0.948)		0.045

* Comparison with age-, sex-, and BMI-matched healthy control group. *^a^*: Adjusted for age and sex. Abbreviations: OR, odds ratio; Coef, coefficient; CI, confidence interval; Ref, reference

## Data Availability

All data generated or analyzed during this study are included in this published article.

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
