# Peer review of "Endocrine and Metabolic Illnesses in Young Adults with Prader–Willi Syndrome"

_jpm, 2022, doi:10.3390/jpm12060858_

Round 1

Reviewer 1 Report

The article titled, Endocrine and metabolic illnesses in young adults with Prader Willi Syndrome has articulated results well from the cohort study. The article has been suggeste

Author Response

We really appreciate your precise summary. Thank you for the feedback.

Reviewer 2 Report

“Endocrine and metabolic illnesses in young adults with Prader-Willi syndrome” was a retrospective cohort study of 68 individuals with Prader-Willi syndrome and compared the prevalence of endocrine and metabolic illnesses with the healthy control group. The study employed simple statistics in order to show the differences. I advise to run a multivariable logistic regression model on 68 patients to evaluate the age- and sex-adjusted association of GH treatment and each one of the outcome variables such as diabetes, hypertension, dyslipidemia, thyroid disease, decreased bone density and sleep apnea.

If the authors are able to add patients’ lifestyle data such as sleep duration and patter, diet pattern and physical activity, the article would be more enlightening and illustrative.

Numerical findings should be presented in the Abstract.

Reviewer 3 Report

Please see the attached review.

Round 2

Reviewer 2 Report

Thanks for the great improvements. Please construct the univariable and multivariable logistic regression models for metabolic syndrome and add the findings to the Table 4-2 as the first row of table. Please add the descriptive findings for metabolic syndrome to the Tables 1, 2 and 4-1 as well. Obviously you need to add the related explanations to the Methods.

Reviewer 3 Report

Please see the attached review.
